# Predatory activity and nematocidal compounds released into liquid culture filtrates as attack strategies of a Mexican strain of *Arthrobotrys oligospora* against *Haemonchus contortus* infective larvae

Héctor Alejandro de la Crúz-Crúz[1,2,3], Rosa Isabel Higuera-Piedrahita[2☯*],
Yazmín Alcalá-Canto[4‡], Alejandro Zamilpa[5‡], Ana Yuridia Ocampo-Gutiérrez[1,2‡],
Luis David Arango-de la Pava[2‡], Gerardo Ramírez-Rico[2‡],
Génesis Andrea Bautista-García[1‡], Gustavo Pérez-Anzúrez[1‡], Agustín Olmedo-Juárez[1‡],
María Manuela Reyes-Estebanez[6‡], Pedro Mendoza de Gives[1☯*]

1 Laboratory of Helminthology, National Centre for Disciplinary Research in Animal Health and Innocuity (CENID-SAI), National Institute for Research in Forestry, Agriculture, and Livestock, INIFAP-SADER, Jiutepec, Morelos, México, 2 Faculty of Professional Studies Cuautitlan, National Autonomous University of Mexico, Cuautitlán, State of México, México, 3 Master's and Doctorate Programs in Health Sciences and Animal Production, Faculty of Veterinary Medicine and Animal Husbandry, National Autonomous University of Mexico, Coyoacán, Ciudad de México, México, 4 Faculty of Veterinary Medicine and Zootechnics, National Autonomous University of Mexico, Coyoacán, Ciudad de México, México, 5 Southern Biomedical Research Center, CIBIS-IMSS, Argentina 1, Col. Centro, Xochitepec, Morelos, México, 6 Department of Environmental Microbiology and Biotechnology, Autonomous University of Campeche, Campeche, México

☯ These authors contributed equally to this work.
‡ These authors also contributed equally to this work.
* pedromdgives@yahoo.com (PMG); rositah_10@cuautitlan.unam.mx (RIHP)

## Abstract

Nematophagous fungi offer a sustainable alternative for controlling nematode infections in small ruminants. The aims were to isolate and characterize both morphological and molecular nematophagous fungi from soil, to assess their predatory activity and the nematocidal activity of their liquid culture filtrates (LCF) against *Haemonchus contortus* infective larvae (L3), and to identify the protease activity of the LCF and mycoconstituents. The isolated and characterized *Arthrobotrys oligospora* was identified using both morphological and molecular techniques, with a similarity of 98%. Additionally, the isolated strain showed 89% phylogenetic similarity in the phylogenetic tree concerning the *Arthrobotrys* order. The *A. oligospora* isolate exhibited 72.06% predatory activity, and the liquid filtrate demonstrated 96.10% nematocidal activity at 100 mg/mL after 48 hours post-exposure against *H. contortus* infective larvae. Regarding enzyme activity, *A. oligospora* showed metalloprotease and cysteine-protease activities, and the zymogram revealed that these activities were higher under acidic conditions (pH 5).

**Data availability statement:** The manuscript contains all raw data required to replicate the results of our study.

**Funding:** This study was supported by the Secretariat of Science, Humanities, Technology, and Innovation (SECIHTI) for the scholarship awarded to Professor Héctor Alejandro de la Cruz (registration number 713914). This research was also supported by the SECIHTI scholarship (Frontier Sciences Project-2023, scholarship number CF-2023-I-2309) awarded to Dr. Pedro Mendoza de Gives. This work also received funding from the Program to Support Research and Technological Innovation Projects from General Directorate of Academic Personnel Affairs of the National Autonomous University of Mexico (UNAM DGAPA-PAPIIT) (grant no. 200324) "Molecular docking study of two lignans, 3-dimethoxy-isoguayacin and norisoguayacin, obtained from Artemisia cina against COX-2," received by Dr. Rosa Isabel Higuera Piedrahita. This study was also supported by the Research Chair Program of the Faculty of Higher Studies Cuautitlán (Cátedra FESC) (grant no. CI2428) awarded to Dr. Rosa Isabel Higuera Piedrahita.

**Competing interests:** The authors have not competing interests.

## Introduction

Gastrointestinal parasitic nematodes (GIN) are among the most significant concerns in the livestock industry worldwide. This has led to a considerable deterioration not only in animal health but also to an enormous economic impact on the livestock industry worldwide [1–3]. In addition, the typical strategy used for ages against these parasitic diseases involves the frequent and continuous administration of chemically synthesized anthelmintic drugs (CSAD), which help mitigate animal damage and indirectly enhance their productivity. However, several disadvantages make this system an overlay. In this regard, the frequent and continuous use of CSAD in animals triggers the development of mutations in one or more genes of the parasites, enabling them to overcome the lethal effect of CSAD and become resistant to these drugs [4–6]. Additionally, the use of such CSADs can lead to residues of these chemicals remaining in meat, milk, and by-products intended for human consumption, posing a potential contaminant risk and a threat to public health [7]. Likewise, CSAD is eliminated by the treated animals through urine and feces, which ultimately reach the soil, where it can affect non-target beneficial microorganisms that comprise the soil microbiome [8] and even impact aquifers, resulting in devastating environmental consequences for water fauna [7]. To summarize, the use and misuse of CSAD can threaten soil microbiota and affect human health. The growing problem of anthelmintic resistance in the parasites, together with other disadvantages, has provoked a bad reputation for the use of these drugs as a unique method of control. This problem requires exploring other alternative methods different from the use of CSAD, such as management systems, i.e., grazing rotation [8,9], grazing alternation of species [10], use of plant/plant metabolites with anthelmintic activity [11–13], vaccines [14,15], biological control using natural nematode enemies such as the nematophagous fungi [16,17]. Moreover, products are derived from the secondary metabolism of nematophagous fungi [18]. Nematode trapping fungi (NTF) are microorganisms found in soil that are considered the primary natural enemies of nematodes in various environments [19]. NTF possesses the capability to be a saprophytic microorganism. However, in the proximity of nematodes, they transform their mycelia into trapping devices designed to capture nematodes, as well as specialized ones that feed on their tissues [16,18,20]. The genus *Arthrobotrys* is one of the most widely studied NTF that has displayed several biological capabilities, including as a bio-regulator of nematode populations [19], cellulose-degrading enzyme activity [21], endophyte activity [20], apoptotic activity enhancer of cancer cells [22], producer of nematocidal secondary metabolites [23], among other essential vital activities. This study was designed to isolate and characterize both morphological and molecular nematophagous fungi from soil and to assess their predatory activity and the nematocidal activity of their liquid culture filtrates (LCF) against Haemonchus contortus infective larvae (L3). Additionally, we investigated the protease activity of LCF.

## Materials and methods

### Location

This study was conducted at the Laboratory of Helminthology, CENID-SAI, INIFAP, in Jiutepec, Morelos, Mexico, and at Laboratory 3, Unit of Multidisciplinary Research, FES-Cuautitlán, UNAM, in Cuautitlán Izcalli, Mexico.

## Sampling site

Soil sampling was conducted at the Sayil archaeological zone, located in the Santa Rosa Municipality of the Puuc Region, Yucatán, Mexico. Fifty grams of soil, 10 cm deep and 30 cm from the trunk of an *Enterolobium cyclocarpum* (Jacq.) leguminous tree. Griseb, also known as the "Pich tree" or "Guanacaste" (Fig 1), was collected. To carry out the sampling, a special permit was processed before the National Service of Agrifood ®Health, Safety and Quality (SENASICA) with an approval date of May 2023.

## Biological material

**Fungal isolation.** The process of fungal isolation was conducted with utmost precision and care. Five milligrams of soil were delicately sprinkled on the surface of water agar plates (3 plates), and a 5 ml aqueous suspension containing 600 specimens of the free-living nematode *Panagrellus redivivus* was added to each plate to enhance the growth of nematophagous fungi [24]. After a week, the surface of the plates was meticulously examined under a stereomicroscope, searching for the formation of trapping devices and trapped nematodes, as well as taxonomic structures typical of nematophagous fungi. Fungi were then transferred to sterile water agar plates, and pure cultures were obtained by transferring to new sterile water agar plates [25].

**Morphological taxonomic identification.** The morphological identification of fungi was performed by observing structures of taxonomic importance under a light microscope (Leica® ZEISS DM6, Wetzlar, Hesse, Germany) at magnifications of 25x, 40x, 75x, and 100x. The length and width of twenty-five conidia and conidiophores were randomly measured, and a range of these measurements was recorded. The presence of branched or unbranched conidiophores,

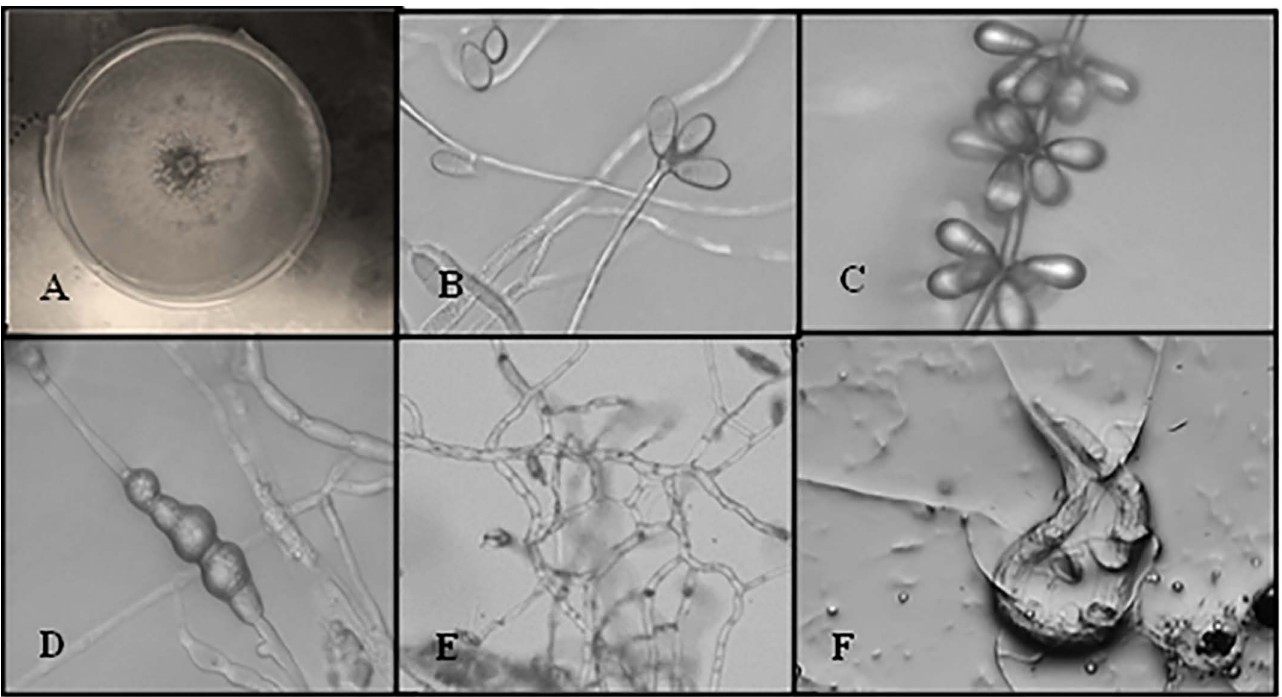

**Fig 1. Aspect of a macroscopic view of a water agar plate containing a 7-day old *Arthrobotrys oligospora* isolate (A); Single conidiophore showing the development of three apical conidia cluster (B); A conidiophore showing the development of repeated conidia cluster proliferation along the conidiophore stem (C); The presence of catenulate chlamydospores (D); three-dimensional adhesive nets (E) and a trapped larva (F).**

the number of conidial septa, the kind of trapping devices, and the presence or absence of chlamydospores were also considered. Our findings were compared with those reported in specialized taxonomic keys to establish the genus and species of our isolate, finally [26,27].

**Molecular identification.** The strain was produced in Czapek-Dox Agar plates for 14 days at room temperature (18–25°C). After this time, abundant cottony mycelium grew on the surface of the plates. Mycelia were collected from culture plate surfaces using a sterile needle and divided into 1.5 ml Eppendorf tubes. DNA extraction was achieved by grinding the mycelia using lysis beads and spinning down at 4000 rpm for 5 minutes. The Wizard® Genomic DNA Purification Kit (Promega, Madison, WI, USA) was used. The DNA genomic quantification was conducted using an IMPLEN spectrophotometer (NanoPhotometer® NP80). DNA amplification was achieved. By PCR using the ITS4 (50-GGAAGTAAAAGTCGTAACAAGG-30) and ITS5 (50- TCCTCCGCTTATTGA-TATGC-30) primers, with a C1000 Touch® Thermal Cycler (Bio-Rad, Hercules, CA, USA). The PCR conditions were established as follows: initial denaturation at 94 °C for 3 minutes; 35 cycles of denaturation at 94 °C for 1 minute, annealing at 42 °C for 90 seconds, and extension at 72 °C for 90 seconds, followed by a final extension stage at 72 °C for 5 minutes. The size of the amplicons was confirmed using a 1.5% agarose gel.

The QIAquick gel extraction kit (QIAGEN) was used, and the procedure was performed with purified products, following the manufacturer's instructions. Genomic DNA was sequenced at the Institute of Biotechnology of the National Autonomous University of Mexico (IBT-UNAM) using an Applied Biosystems sequencer. The sequences were aligned using the NCBI BLAST program (www.ncbi.nlm.nih.gov/blast/) accessed on April 30, 2024 (Basic Local Alignment Search Tool).

## Nematodes

**Panagrellus redivivus.** A strain of the free-living nematode *P. redivivus* was commercially obtained as a commercial food for fish from Heber Martinez Pateiro, Mexico. This strain was reproduced by culturing sterile wet oat flakes in plastic bowls covered with gauze, incubating at room temperature (26–28°C) for seven days. After this period, nematodes were recovered from bowls using the Baermann funnel technique for 24 h [26,28]. Recovered nematodes were sieved through a 100 μm mesh and resuspended in sterile distilled water.

***Haemonchus contortus* infective larvae (L3).** A four-month-old male lamb, artificially infected with 5,000 L3 *H. contortus* infective larvae (FESC strain) previously dewormed, was used as a donor animal for nematode eggs. After 18 days of the pre-patent period, feces were directly collected from the rectum of this animal to prepare fecal cultures in plastic bowls by the modified Corticelli-Lai technique [29]. The *H. contortus* strain (FESC strain) It has been isolated and maintained at the Faculty of Higher Studies Cuautitlán, and has also been characterized with heterozygous genes for benzimidazole resistance [30].

The sheep donor for *H. contortus* strain was maintained under controlled conditions following the Norma Oficial Mexicana (Official Mexican Standard) with official rule number NOM-052-ZOO-1995 (http://www.senasica.gob.mx). Additionally, the experimental protocol was approved by the Internal Committee for Care and Use of Experimental Animals (CICUAE-FESC) at Facultad de Estudios Superiores Cuautitlán, Universidad Nacional Autónoma de México, under protocol number C24_25 (S1 Fig).

**Predatory activity assessment.** The nematode/fungus confrontation was performed in plastic petri dishes containing water agar. Ten plates (60 *15 mm) were inoculated with the mycelia of the fungal isolate and incubated for seven days at room temperature, 25–28°C. The other ten plates containing only the medium (without fungus) were used as controls. One milliliter of an aqueous suspension containing 600 *H. contortus* infective larvae was deposited on the surface of each plate of both groups. The whole plates were incubated for ten days at the temperature mentioned above. After incubation, the agar from each plate was removed and placed on a Baermann funnel system to recover non-trapped larvae from the treated group and whole larvae from the control group plates. The larvae quantification was achieved using the same procedure previously described. Results were based on the means of recovered and standard deviation larvae from both

experimental groups. The reduction percentage of recovered larvae attributed to the predatory effect of the fungus was estimated using the ABBOTT formula [31]:

$$PA\% = \frac{(RLc - RLt)}{RLc} * 100$$

Where:
PA% =Predatory activity (percentage).
RLc=Mean of recovered larvae in plates with no fungi.
RLt=Mean of recovered larvae in plates with fungi.

## Statistical analysis

The ANOVA test following by Tukey test, which analyzed data, proved that the means of recovered larvae in the two experimental groups were the dependent variables. The SAS statistical package, SAS 9.0, was used.

**Liquid filtrate obtaining.** The fungal isolate grew on water agar plates for seven days at room temperature (25–28°C). A small cylinder plug (1 x 1 cm) was taken from the fungal cultures and deposited in 500 mL flasks containing 150 mL of Czapek-Dox broth®. Flasks were incubated under static conditions at the same temperature described above for 21 days. After this period, the liquid from the cultures was filtered using three different filtration systems: a coffee filter, Whatman paper No. 4, and finally, a 0.45 µm and 0.22 µm filter system (KIMBLE® ULTRA-WARE®, Mainz, Germany). Liquid culture filtrates were frozen using a blast freezer (BIOBASE®, Shangdong, China). After the freezer material was lyophilized using a conventional lyophilizer (LABCONCO®, Kansas, USA). Lyophilized LCFs were kept at room temperature (25–28°C) until use.

**Assessment of the lethal activity of liquid culture filtrates against *Haemonchus contortus* larvae.** The nematode/LCF confrontation was conducted in 96-well plastic microtitering plates. Three different concentrations of LCF were established: 25, 50, and 100 mg/mL. A hundred microliters of each concentration and 100 mL of an aqueous suspension containing 500 *H. contortus* infective larvae were deposited in the corresponding well (n = 4). Czapek-Dox broth® (fungus-free), 10% ivermectin, and sterile distilled water were used as controls. The plate was incubated at room temperature (25–28°C) for 72 h. After incubation, the number of dead and live larvae in each well of the entire experiment was counted by taking five 40-ml aliquots from each well. The results were considered when calculating the means of dead and live larvae in all the treatments (three concentrations and their controls). The mortality percentages attributed to the effect of LCF were estimated using the ABBOTT formula, as follows:

$$LMP = MRDL/MRLL + MRDL * 100$$

Where:
LMP=Larval mortality percentage.
MRDL Mean of recovered dead larvae.
MRLL Mean of recovered live larvae.

**Statistical design.** A completely random analysis was performed using an ANOVA analysis, with dead and live larvae as the dependent variables. The Tukey complementary method was used. The SAS statistical package, SAS 9.0, was used.

**Microscopic analysis.** A set of microphotographs was recorded of both the predatory activity and the effect of LCF on H. contortus larvae to visualize the fungal predatory activity and the significant damage exerted by LCF on the nematodes. Photographic material was captured using a Leica DM6 B compound microscope (Wetzlar®, Germany).

**Zymography assays.** Proteolytic activities were determined according to the method described by Ramírez-Rico et al. [32]. The technique utilized 10% polyacrylamide gels copolymerized with 0.2% bovine gelatin (Sigma-Aldrich®). The

protein concentration was adjusted to 20 μg of LCFs of *Arthrobothrys oligospora* for electrophoresis. Electrophoresis was performed at 100 V for 2 hours at 4°C, without treating the samples with β-mercaptoethanol or boiling them. After electrophoresis, the gels were washed twice with a 2.5% (v/v) Triton X-100 solution (Sigma-Aldrich®) for 30 minutes and then incubated overnight at 37°C with different buffer solutions: 100 mM sodium acetate, 2 mM CaCl2 (pH 5.0); 100 mM Tris, 2 mM CaCl2 (pH 7.0); and 100 mM Na2CO3-NaHCO3, 2 mM CaCl2 (pH 9.0) (all from Sigma-Aldrich®). Finally, the gels were stained with 0.5% (w/v) Coomassie Brilliant Blue R-250 (Bio-Rad®, Feldkirchen, Germany). Proteolytic activities were detected as clear bands against a blue background. The gels were washed until the proteolytic bands were visible.

**Characterization of proteolytic activity using protease inhibitors.** The LCF of *A. oligospora* was pre-incubated for 1 hour at 22°C with different inhibitors under constant agitation for the protease inhibition assays, following the technique described by Ramírez-Rico et al. [32]. The inhibitors were used at the following concentrations: 10 mM p-hydroxymercuri benzoate (pHMB) for cysteine proteases; 5 mM phenylmethyl-sulfonyl fluoride (PMSF) for serine proteases; and 5 mM EGTA, 5 mM EDTA, or 10 mM o-phenanthroline for metalloproteases (all obtained from Sigma-Aldrich®). Samples were loaded onto 10% SDS-PAGE gels copolymerized with 0.2% bovine gelatin (Sigma-Aldrich®) and subjected to electrophoresis at 4°C in an ice bath at 100 V for 2 hours. Subsequently, the gels were washed twice with 2.5% (v/v) Triton X-100 solution (Sigma-Aldrich®) for 30 minutes, incubated overnight with 100 mM Tris-2 mM CaCl2 (pH 7.0), and stained as previously described.

## Results

### Traditional morphological taxonomy

Macroscopically examining the plates containing the fungal isolate growing on water agar showed a whitish, cottony growth, forming concentric mycelial rings (Fig 1A). The microscopic observation of the fungus revealed the presence of erect and straightforward conidiophores with initial apical conidial cluster formation (Fig 1B). In older cultures, several conidial clusters formed along the conidiophores (Fig 1A). Additionally, chlamydospores, three-dimensional adhesive nets, and trapped nematodes were observed (Fig 1D–1F).

Table 1 shows the measurements of taxonomic importance for nematophagous fungi, i.e., the length and width of conidia and the length of conidiophores. After comparing the structures and measurements with those described in specialized taxonomic keys, we concluded that our isolate corresponded to *Arthrobotrys oligospora*.

### Molecular taxonomy

The analysis of fungal DNA sequences, followed by alignment and comparison to other isolates reported in the National Center for Biotechnology Information, determined that our isolation corresponds to *A. oligospora*, confirming the results obtained from the traditional morphological analysis. Data on query coverage, similarity percentages, and GenBank accession numbers are presented in Table 2.

Subsequently, the phylogenetic tree (Fig 2) showed that the evolutionary relationships between taxa were determined using the Maximum Likelihood method. This analysis included 43 nucleotide sequences corresponding to the

**Table 1. Mean and range of 25 conidia and conidiophores measurements and characteristics observed under a light microscope.**

| Characteristic | N | Mean (μm) | Standard deviation (μm) |
|---|---|---|---|
| Conidia length | 25 | 23.19 | 21.28-25.74 |
| Conidia width | 25 | 12.19 | 10.24-14.88 |
| Conidiophore | 26 | 268 | 220-230 |
| Chlamydospores | | Present | |
| Type of traps | | Adhesive nets | |

**Table 2. Similarity and coverage of the obtained sequence after comparison with reported sequences in the GenBank–NCBI database, using the partial ITS1, 5.8S, and ITS2 regions.**

| Strain | Query cover % | Similarity % | GenBank accession number |
|---|---|---|---|
| Orbilia oligospora | 99 | 98.67 | OM066025.1 |
| Orbilia oligospora | 99 | 98.67 | OM065997.1 |
| Orbilia oligospora | 98 | 98.95 | KJ938573.1 |
| Arthrobotrys oligospora | 97 | 99.09 | MZ427472.1 |
| Orbilia oligospora | 98 | 98.80 | OM066039.1 |

*Arthrobotrys, Dactylellina*, and *Drechslerella* genera. All ambiguous positions were removed for each pair of sequences using the pairwise removal option. The final dataset consisted of 522 base pairs. The phylogenetic analyses were conducted in IQTtree® (version 2.3.6). The best substitution model (SYM+I+G4) was calculated using JModelTest (version 2.1.10).

### Predatory activity assessment

Table 3 shows the fungal predatory activity. The NF was divided into Group 1 (larvae and fungus) and Group 2 (larvae only). Additionally, the mean number of *H. contortus* larvae recovered after ten days of fungal predation on the nematode, as well as the percentage reduction in larvae attributed to its nematocidal capacity, is also observed.

### Lethal activity of liquid culture filtrates against *Haemonchus contortus* larvae

Table 4 shows the percentage of *H. contortus* infective larvae that died after exposure to *A. oligospora* filtrates at concentrations of 100, 50, and 25 mg/mL. The mortality rate decreased from 90.10% to 36.35%, and the filtrate exhibited a concentration-efficacy relationship after 48 hours of exposure.

### Microscopic analysis

Fig 3 shows the damage observed in *H. contortus* infective larvae exposed to *A. oligospora* filtrates (100 mg/mL) after 48 hours. Larvae exhibited changes in their internal structures, particularly with the loss of intestinal cell architecture and a remarkable degree of tissue disorganization. Additionally, a notable loss of movement in the larvae was observed, eventually resulting in them remaining motionless. At lower concentrations, no cell damage was observed in the infective larvae.

### *Arthrobotrys oligospora* proteolytic activities and their stability over a wide pH range

To evaluate the proteolytic activity present in the *A. oligospora* filtrate, we performed zymography assays and evaluated the activity at different pHs. Our results revealed several proteolytic activities (Fig 4). At pH 5, three highly active proteases are mainly observed with molecular weights of 75, 90, and 140 kDa, and a less intense activity of 250 kDa (Fig 4a). At pH 7, a higher number of proteolytic activities were found compared to pH 5, with molecular weights of 37, 45, 55, 75, 90, 100, 140, 155, and 250 kDa (Fig 4b). In the case of pH 9, the same proteases are observed as in pH 7, but the high molecular weight activities between 75 and 140 kDa show greater proteolytic activity (Fig 5c).

### *A. oligospora* secretes metallo and cysteine proteases

The present study demonstrates that *A. oligospora* exhibits significant proteolytic activity, which can be modulated with different pH conditions. To determine what type of proteases *A. oligospora* secretes, we performed proteolytic inhibition assays with different inhibitors and subsequently performed zymograms.

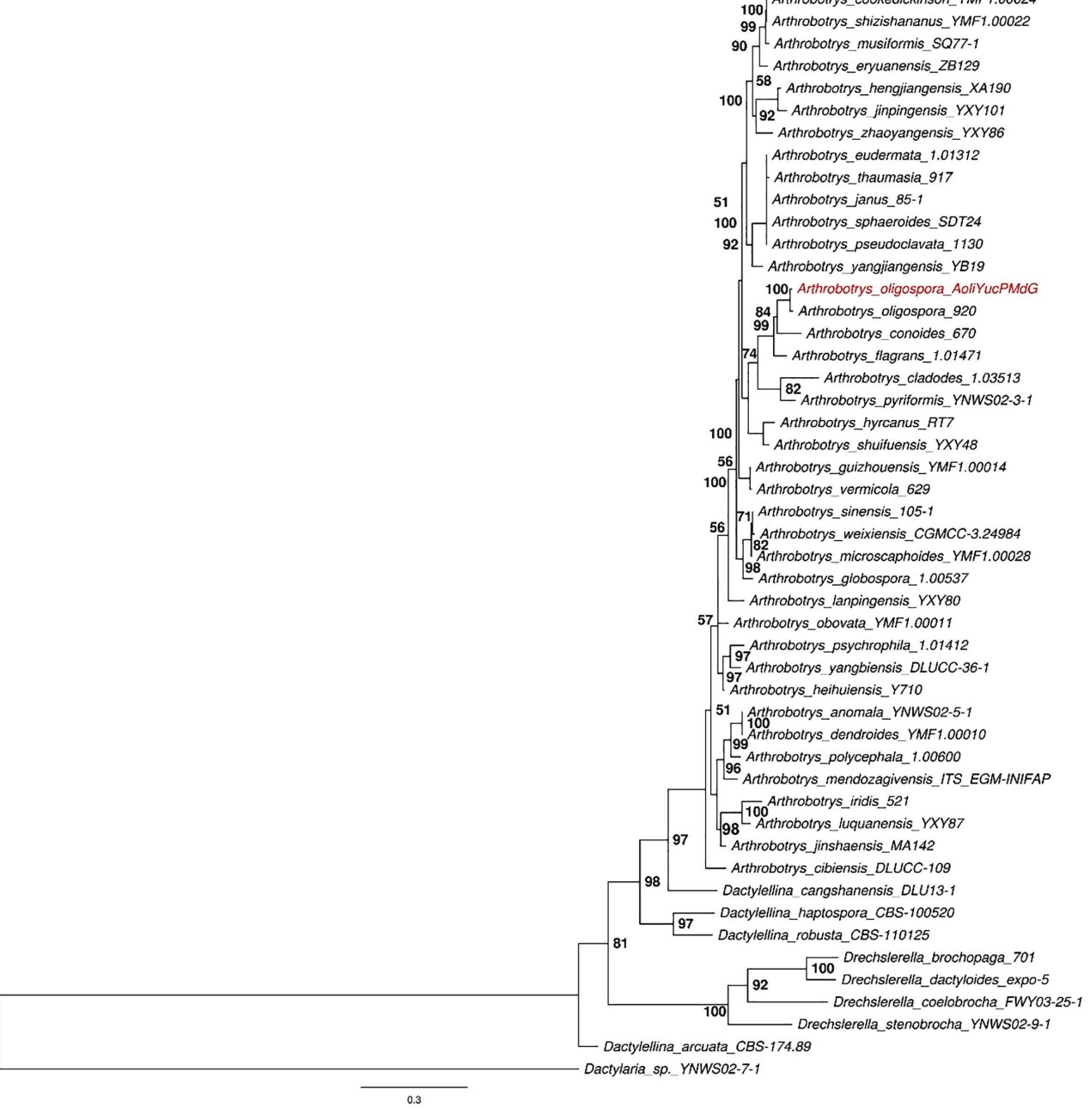

**Fig 2. Phylogenetic tree obtained by maximum likelihood using an ultrafast bootstrapping with 10000 replicates.** Our strain, *Arthrobotrys oligospora* (NCBI: isolate PP741577), is highlighted in red. Bootstrap values are shown in bold.

The results revealed that all compounds tested in this study inhibited low molecular weight proteases (37–55 kDa) (Fig 5, lanes 2–6), compared to the control sample without inhibitors (Fig 5, lane 1). However, when we used EDTA, we observed a complete inhibition of 75 kDa activity and a partial inhibition of high molecular weight proteases (>75 kDa) (Fig 5, lane 2). EGTA partially inhibited proteases from 75 kDa onwards (Fig 5, lane 3). However, the inhibition results with

**Table 3. Mean of *Haemonchus contortus* infective larvae recovered from *Arthrobotrys oligospora* water agar plates after ten days of confrontation (p<0.05).**

| Group | Mean of recovered larvae ±SD | Reduction of larvae % |
|---|---|---|
| Larvae/fungus interaction | 117±35[a] | 72.06 |
| Control | 419±76[b] | – |

**Table 4. Percentage of *H. contortus* infective larvae mortality after exposure to *A. oligospora* culture filtrates for 48 hours.**

| Yield extraction | | | | | | |
|---|---|---|---|---|---|---|
| Group | 100mg/mL | | 50mg/mL | | 25mg/mL | |
| | Dead/Total | Mortality (%) | Dead/Total | Mortality (%) | Dead/Total | Mortality (%) |
| *A. oligospora* filtrate | 96-107 | 90.10 [a] | 91-120 | 75.54 [b] | 37-101 | 36.35 [c] |
| Ivermectin | 100 | 100 [a] | 100 | 100 [a] | 100 | 100 [a] |
| Distilled water | 0 | 0 | 0 | 0 | 0 | 0 |

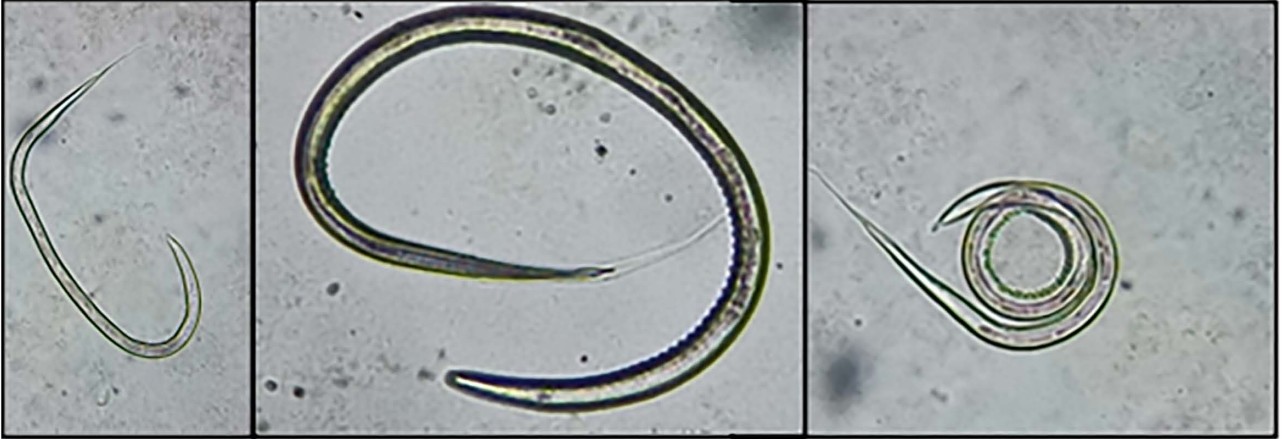

**Fig 3. Photographs showing internal morphological changes at the intestinal cells of *H. contortus* infective larvae exposed to *A. oligospora* filtrates (100 mg/mL) after 48 hours.**

pHMB and mainly with phenanthroline showed a complete inhibition of all proteolytic activities of *A. oligospora* filtrates (Fig 5, lanes 4 and 6). We did not observe inhibition of high molecular weight proteases when we used PMSF (Fig 5, lane 5, S2 Fig).

## Discussion

### Traditional morphological taxonomy

Nematophagous fungi share specific general characteristics across different genera and species of nematode-trapping fungi. These morphological similarities can sometimes complicate traditional taxonomic identification. For instance, several species within the *Arthrobotrys* genus appear quite similar at first sight. However, differences such as the type of conidiophore (whether single or branched), conidium shape and size, trapping devices, and the presence or absence of chlamydospores enable accurate species differentiation.

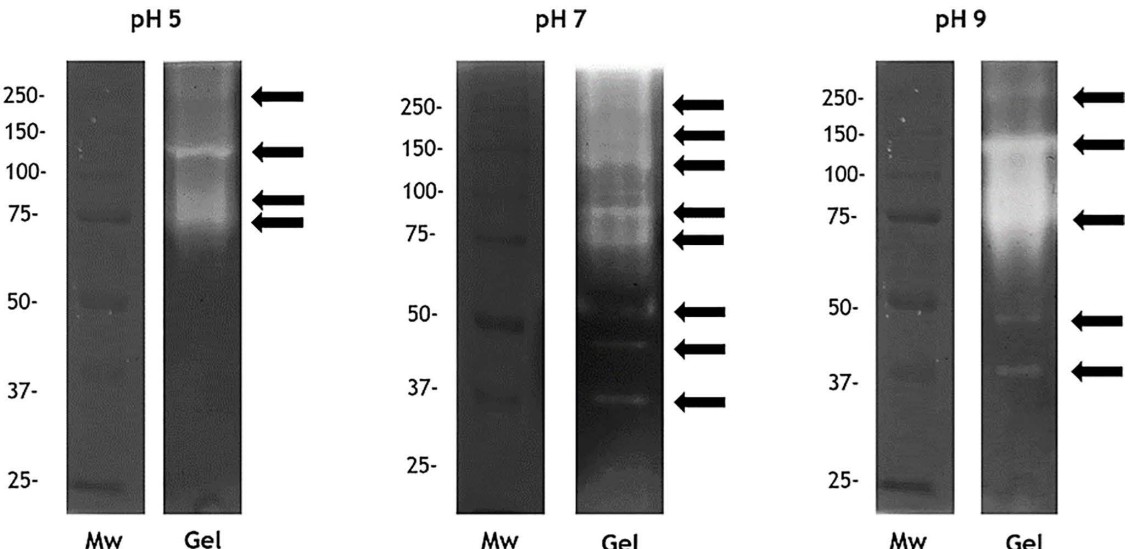

**Fig 4. Zymograms of the liquid culture filtrate of *A. oligospora*.** Twenty micrograms of proteins derived from the culture filtrates of *A. oligospora* were loaded onto 10% polyacrylamide gels copolymerized with 0.2% bovine gelatin. The following buffers were evaluated to determine the activation pH: (a) pH 5.0, (b) pH 7.0, and (c) pH 9.0. In the first lane of each pH evaluated, the reference molecular weight marker (Mw) is observed. The arrows indicate the proteolytic activities evaluated at each pH.

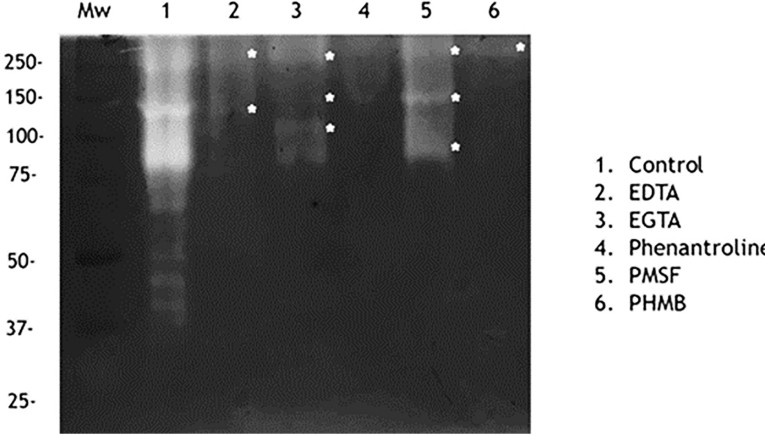

**Fig 5. Inhibition of the proteolytic activity of *A. oligospora* filtrates through zymography.** We performed an incubation of the filtrates before electrophoresis, using the following inhibitors: EDTA (lane 2), EGTA (lane 3), and phenanthroline (lane 4) for metalloproteases; phenylmethylsulphonyl fluoride (PMSF) (lane 5) for serine proteases, and p-hydroxy-mercury-benzoate (pHMB) (lane 6) for cysteine proteases. The sample without inhibitors was used as an experimental control lane (lane 1).

Our isolate produces single, unbranched, erect conidiophores, a characteristic observed in various nematode-trapping fungi, including *Arthrobotrys oligospora, A. musiformis, A. conoides, A. arthrobotryoides, A. dactyloides*, and *A. javanica* [33]. In addition to the type of conidiophores, these species exhibit differences in other traits that help inform our taxonomic diagnosis.

Our isolate developed three-dimensional adhesive nets as trapping devices, a feature also seen in *A. oligospora, A. conoides, A. musiformis*, and *A. arthrobotryoides*. However, *A. dactyloides* is distinguished by producing constricting rings

[34], while *A. sinensis* employs a combination of trapping devices, including both three-dimensional adhesive nets and constricting rings [35].

While *A. oligospora* and *A. conoides* share the characteristics of single and erect conidiophores, the stems of the conidiophores in *A. oligospora* are larger compared to those in *A. conoides*. *Additionally, A. conoides* has smaller conidia, measuring 8.4 x 11.8 μm, while *A. oligospora's* conidia are larger, measuring an average of 23 x 12.1 μm [36]. A summary of the main morphological characteristics of five species within the *Arthrobotrys* genus is provided in Table 5.

## Molecular taxonomy

The phylogenetic tree was constructed using 43 sequences of related nematophagous fungi, which were previously reported in the NCBI database. This analysis revealed that our sequence is closely related to *A. oligospora*_920, a relationship that is strongly supported by a bootstrap value of 100. Interestingly, our isolate also shows a close phylogenetic relationship with *A. conoides*_670, which is found in the same clade. As we noted in the morphological taxonomic identification, these two species are highly similar and share specific characteristics, such as the development of single, erect conidiophores. Both species also produce three-dimensional adhesive nets as trapping devices. Additionally, another species, *A. flagrans* (or *Duddingtonia flagrans*), shares some morphological similarities with these two *Arthrobotrys* species, including the formation of conidia clusters and the same type of trapping devices [42]. The phylogenetic analysis revealed that this species branches off in a distinct part of the evolutionary tree yet remains closely related to others. Beyond our examination of the molecular data confirmed the findings from our morphological analysis. A recent study focused on investigating the functions of mitophagy, conidiation, stress response, and pathogenicity in *A. oligospora. It* revealed that two isolates of *A. oligospora* homologs resulted in a close relationship with orthologs from other nematophagous fungi, including *D. flagrans* [43]. In another study, focused on determining the phylogenetic relationship of *D. flagrans* with other fungi, it was revealed that both *D. flagrans* and *A. oligospora* are grouped in the same clade, indicating that they share the same common ancestor [44].

## Predatory activity assessment

The isolate obtained in the present study exhibited significant predatory activity, with a rate higher than 70% against the infective larvae *H. contortus*, comparable to the predatory behavior shown by other isolates from other countries (Table 6).

**Table 5. Key morphological characteristics of taxonomic importance exhibited by various species of the nematophagous fungus in the *Arthrobotrys* genus.**

| Species | Trap Structure | Hyphal Diameter (μm) | Conidia (Size & Shape) | Conidiophore | Chlamydospores | Reference |
|---------|----------------|----------------------|------------------------|--------------|----------------|-----------|
| *A. oligospora* | 3D adhesive nets (most common) | 2–5 | Oval/ellipsoidal, 15–30 × 6–12 μm | Simple, erect, bottle-shaped | Present (survival under stress) | [37] |
| *A. conoides* | 3D adhesive nets | 2–4 | Oval, 12–25 × 5–10 μm | Simple, bottle-shaped | Rare or absent | [38] |
| *A. musiformis* | 3D adhesive nets | 2.5–6 | Elongated, 20–35 × 5–10 μm | Simple, elongated phialides | Present in some strains | [39] |
| *A. dactyloides* | Constricting rings (less common than nets) | 2–4 | Oval-cylindrical, 15–30 × 5–10 μm | Simple, short phialides | Absent | [40] |
| *A. sinensis* | Adhesive nets + constricting rings (strain-dependent) | 2–5 | Oval, 10–25 × 4–8 μm | Simple, bottle-shaped | Rare (reported in some strains) | [41] |

**Table 6. Comparative table showing the in vitro predatory activity of different *Arthrobotrys oligospora* isolates against *Haemonchus contortus* infective larvae.**

| Predated parasite | Predation percentage | Country | Reference |
|---|---|---|---|
| *Haemonchus contortus* | 91% | Mexico | [45] |
| *H. contortus* | 78.82% | China | [46] |
| *H. contortus* | 100% | India | [47] |
| *H. contortus* | 45.14% | Mexico | [48] |
| *H. contortus* | 79.7% | Iran | [49] |
| Gastrointestinal parasitic nematodes (L3) | 71.10% | Colombia | [50] |

The practical application of nematophagous fungi for controlling parasitic nematodes in ruminants involves the oral administration of spores from selected fungal isolates. After passing through the animals' digestive tracts, these spores are excreted in the feces. Once in the feces, the spores germinate in situ, initiating a morphogenesis process that transforms their mycelia into trapping devices [44]. The most commonly used genus of nematode-trapping fungus for controlling ruminant parasitic nematodes is Duddingtonia flagrans, which is known for its ability to develop a large number of chlamydospores. These chlamydospores are more resistant than simple spores and can survive after passing through the gastrointestinal tract of animals [42]. Nevertheless, *A. oligospora* spores have also been reported to survive through the gastrointestinal tract of small ruminants, and after reaching the feces, they germinate and exert their predatory activity [51]. Other species have also demonstrated their ability to survive their passage through the digestive process in ruminants while maintaining their predatory activity (Table 7). The *A. oligospora* isolate used in this study could be a promising candidate for further assessment as a potential biocontrol agent in future experiments.

## Lethal activity of liquid culture filtrates against *Haemonchus contortus* larvae

The larval mortality of *H. contortus*, attributed to the lethal effects of the liquid culture filtrates of *A. oligospora*, showed significant results. Mortality rates ranged from 36.35% at the lowest concentration (25 mg/mL) to over 90% at the highest concentration (100 mg/mL). A clear dependence on concentration was observed. This finding is noteworthy, as the highest percentage of mortality was observed at the highest concentration of the liquid filtrates. It is essential to consider that such a high larval mortality rate was achieved using a liquid culture filtrate that may contain various compounds, which could mask the effects of the molecules responsible for this mortality. Therefore, we would expect to see greater activity with much lower concentrations of the purified compound. In future studies, we plan to elucidate the compound(s) responsible for the nematocidal activity, which could lead to the discovery of a natural compound with nematocidal properties. Different authors report the use of liquid culture filtrates showed un table 8.

**Table 7. Comparative table showing some nematode-trapping fungi isolates in different *in vivo* studies to assess their potential use in reducing the infective larval population of gastrointestinal parasitic nematodes in feces.**

| Fungus | Specie | Passage through the digestive tract | Percentage reduction | Doses | Authors |
|---|---|---|---|---|---|
| *Arthrobotrys oligospora* | Sheep | + | 99.51% | $5 \times 10^5$ spores/kg | [51] |
| *A. oligospora* | Sheep | + | 95.8% | $9 \times 10^7$ spores/sheep (in corn) | [52] |
| *Monacrosporium thaumasium (A. thaumasium))* | Bovine | + | ≈100% | 20 g of pellets 2x/week (6 months) | [53] |
| *Arthrobotrys robusta* | Bovine | + | 70.45% | $2 \times 10^6$ conidia 2x/week (4 months) | [42] |
| *Duddingtonia flagrans* | Sheep | + | 89.8% | $1 \times 10^6$ clamydospores/kg | [54] |
| *D. flagrans* | Sheep | + | 97.4% | $1 \times 10^6$ clamydospores/kg | [55] |

**Table 8. Percentage of larval mortality of liquid culture filtrates of different isolates of nematode-trapping fungi against infective larvae of parasitic nematodes.**

| Fungus | Liquid medium | Target parasite | Mortality percentage | Authors |
|---|---|---|---|---|
| *Paecilomyces lilacinus* | Czapeck-Dox | *Meloidogyne incognita* | 78.28% (12 h exposure) | [56] |
| *Volutella citrinella* | Potato dextrose | *Aphelenchoides besseyi* | 100% (72 h exposure) | [57] |
| | | *Bursaphelenchus xylophilus* | 100% (72 h exposure) | |
| | | *Ditylenchus destructor* | 55.63% (72 h exposure) | |
| *Arthrobotrys oligospora* | Czapek-Dox | *Haemonchus contortus* (L3) | 83% (72 h exposure) | [58] |
| *Arthrobotrys oligospora* | Potato dextrose | *H. contortus* (L3) | Data not specified (lower than Cz-DB) | |
| *Paecilomyces lilacinus* (MICLAB009) | Minimal medium broth (crude macerate) | *Ancylostoma* spp. eggs | 68.43% (24 h exposure) | [59] |
| *Trichoderma harzianum* | Minimal medium broth (crude macerate) | *Ancylostoma* spp. eggs | 56.43% (24 h exposure) | |
| *Duddingtonia flagrans* | Custom liquid medium (glucose, casein, salts) | *Angiostrongylus vasorum* (L1) | 53.5% (24 h), 71.3% (48 h) | [60] |

## Microscopic analysis

The examination of larvae exposed to liquid culture filtrates of *A. oligospora* under a microscope revealed significant damage, particularly to the internal structures of the larvae. Notably, the intestinal cell architecture was lost only in the larvae that were exposed to the filtrates, while larvae in the control group that were not exposed showed no such damage. This fact suggests that compounds derived from the fungal secondary metabolism present in the liquid filtrates may be responsible for the degradation of the intestinal cells. Exposed larvae gradually reduced their movements until they finally remained motionless until death. Future studies utilizing confocal microscopy, along with immunolocalization markers and fluorophores, could provide more detailed information about the specific sites where the bioactive compounds affect the internal tissues of the larvae.

The mechanism by which *A. oligospora* filtrates induces the observed damage to the larval intestinal cells, we propose that this is likely a multi-faceted process involving both enzymatic degradation and the direct toxic activity of metabolites. The proteolytic enzymes, particularly the metallo- and cysteine proteases identified in our zymograms, could initiate the process by degrading the proteinaceous components of the larval cuticle, compromising its structural integrity and creating pathways for internal entry [61,62]. Once this barrier is weakened, low molecular weight nematocidal metabolites, such as the polyketide-terpenoid hybrids known to be produced by *A. oligospora* [23], could passively diffuse into the pseudocoelom. Their lipophilic nature would allow them to cross cell membranes and act directly on internal organs, including intestinal cells, disrupting their architecture and function. Concurrently, the oral route cannot be discounted; larvae likely ingest the filtrate, introducing proteases and metabolites directly into the digestive tract. Ingested enzymes could damage the intestinal epithelium from the lumen, while absorbed metabolites could exert a systemic toxic effect. Therefore, we suggest a synergistic and sequential model: proteolytic attack facilitates the internalization of toxic compounds, and the combined action of ingested enzymes and absorbed metabolites leads to the severe internal disorganization and eventual death of the larva. Future studies using immunohistochemical localization of these proteases and metabolites within the nematode would be invaluable to confirm this proposed model.

## Zymography assays

*Arthrobotrys oligospora* has been reported to produce extracellular proteases that are involved in its nematicidal effect [61,63,64]. Traps are essential components for nematophagous fungi, and their alteration decreases their predation

efficiency. *A. oligospora*, belonging to this group, has been reported to produce extracellular serine proteases that degrade the nematode cuticle, promoting fungal penetration and colonization [51,65,66].

In this work, *A. oligospora* secretes a large amount of cysteine and metalloproteases of different molecular weights, with proteolytic activity over a wide pH range. Tunlid and Jansson [64] reported in 1991 that *A. oligospora* secretes proteases, which are inhibited by serine protease inhibitors, aspartate protease inhibitors, and cysteine protease inhibitors. They also observed a significantly decreased immobilization of nematodes captured by the fungus after incubation with serine protease inhibitors, PMSF, antipain, or chymostatin, or the metalloprotease inhibitor phenanthroline [65]. This finding is consistent with the results obtained, which showed inhibition of proteolytic activity with cysteine and metalloprotease inhibitors.

The study of proteases secreted by *A. oligospora* has focused on two low-molecular-weight serine proteases. Tunlid et al. [65] purified and characterized an extracellular serine protease of 35 kDa (PII) [61,62]. On the other hand, Junwei et al. [67] purified the serine protease XAoz1 with a molecular weight of 50 kDa. Both proteases participate in the nematicidal effect of nematode-trapping fungi. In this study, we detected low molecular weight proteolytic activities, which coincided with the molecular weights reported in both studies, and also found inhibition with PMSF.

Chitinases are hydrolytic enzymes involved in the crucial digestion of nematode cell walls, particularly during egg parasitism by nematophagous fungi [68]. It has been reported that *A. oligospora* secretes chitinases with molecular weights greater than 100 kDa [69]. The serine protease PII can degrade cuticle, where chitin is present [70]. Therefore, the proteases secreted by *A. oligospora* could also have chitinase activity. Furthermore, the pHMB inhibitor is capable of inhibiting glycosidases to which chitinases belong [70], so the high-molecular-weight proteases in this study may possess this enzymatic activity, which aligns with our proteolytic inhibition results.; however, we did not evaluate the ability of these proteases to degrade chitin. It would be interesting to evaluate in future assays, in addition to their purification and characterization. These proteases could also contribute to the pathogenicity of this fungus, which is involved in the nematicidal effect that traps nematodes, and demonstrate that *A. oligospora* secretes different types and classes of proteases.

### Characterization of proteolytic activity using protease inhibitors

The enzymatic inhibition assay performed reveals the nature of the proteases present, demonstrating the activity of low molecular weight enzymes (37–55 kDa) that were completely inhibited by phenanthroline and pHMB, indicating their dependence on metals and thiol groups, characteristic of metalloproteases and cysteine proteases. Additionally, partial inhibition was observed with EDTA, suggesting the presence of calcium-dependent enzymes. At the same time, PMSF did not affect the high molecular weight proteases, ruling out significant involvement of serine proteases in this range. The results obtained partially contrast with those of Tunlid et al. [61], who identified serine proteases as the main contributors to the nematicidal activity in *A. oligospora*. This discrepancy could be due to differences in strains or culture conditions, highlighting the enzymatic variability among fungal isolates. Overall, the data emphasize the complexity of *A. oligospora's* enzymatic arsenal and its potential for the development of biocontrol agents based on specific enzymes. This finding aligns with previous studies, such as those by Wang et al. [52], which underscores the key role of extracellular proteases in nematode cuticle degradation. The presence of these enzymes in the LCF could explain the high mortality of Haemonchus contortus observed in the study, as they facilitate the penetration and digestion of parasitic tissues by nematophagous fungi

### Conclusion

This study comprehensively demonstrates the high biocontrol potential of a Mexican strain of *Arthrobotrys oligospora* against the parasitic nematode *Haemonchus contortus*. The fungus employs a dual attack strategy: a direct predatory mechanism with a high trapping efficiency (72.06%), and the release of potent nematocidal compounds into its liquid culture filtrate (LCF), which caused up to 96.10% larval mortality.

The findings significantly advance the field by characterizing the enzymatic arsenal behind this activity, identifying a suite of pH-stable metallo- and cysteine proteases as key virulence factors. Furthermore, the chromatographic profiling suggests that secondary metabolites, including lignans and an ellagic acid derivative, may contribute to the observed lethal effect.

This multi-faceted mode of action—combining physical trapping, enzymatic degradation, and bioactive metabolite production—makes this Mexican *A. oligospora* isolate a highly promising and sustainable candidate for the development of novel biocontrol agents against livestock nematodes, offering a viable strategy to counteract the growing issue of anthelmintic resistance.

## Supporting information

**S1 Fig. Ethical approval.**
(PDF)

**S2 Fig. Gel of the zymograms.**
(PDF)

## Acknowledgments

This study formed part of the PhD thesis work of MVZ Héctor Alejandro de la Cruz Cruz at the National Autonomous University of Mexico (UNAM), Mexico, under the direction of Pedro Mendoza de Gives and the tutelage of Yazmin Alcalá Canto and Alejandro Zamilpa.

For MSc Elvia Adriana Morales Hipólito and Hugo Cuatecontzi Flores, who supported the UPLC technique at Facultad de Estudios Superiores Cuautitlán-UNAM, México.

**Institutional review board statement:** The sheep donor for *Haemonchus contortus* strain was maintained under controlled conditions following the Norma Oficial Mexicana (Official Mexican Standard) with official rule number NOM-052-ZOO-1995 (http://www.senasica.gob.mx, accessed on 7 May 2024) and the Ley Federal de Sanidad Animal (Federal Law for Animal Health) DOF 07-06-2012 were strictly applied (https://www.gob.mx/cms/uploads/attachment/file/118761/LFSA.pdf, accessed on 7 May 2024). Additionally, the experimental protocol was approved by the Internal Committee for Care and Use of Experimental Animals (CICUAE-FESC) at Facultad de Estudios Superiores Cuautitlán, Universidad Nacional Autónoma de México, under protocol number C24_25

## Author contributions

**Conceptualization:** Héctor Alejandro de la Crúz-Crúz, Rosa Isabel Higuera-Piedrahita, Yazmín Alcalá-Canto, Alejandro Zamilpa, Pedro Mendoza de Gives.

**Formal analysis:** Héctor Alejandro de la Crúz-Crúz, Alejandro Zamilpa, Agustín Olmedo-Juárez, Pedro Mendoza de Gives.

**Funding acquisition:** Pedro Mendoza de Gives.

**Investigation:** Héctor Alejandro de la Crúz-Crúz, Yazmín Alcalá-Canto, Alejandro Zamilpa, Ana Yuridia Ocampo-Gutiérrez, Luis David Arango-de la Pava.

**Methodology:** Héctor Alejandro de la Crúz-Crúz, Rosa Isabel Higuera-Piedrahita, Yazmín Alcalá-Canto, Ana Yuridia Ocampo-Gutiérrez, Luis David Arango-de la Pava, Gerardo Ramírez-Rico, Génesis Andrea Bautista-García, Gustavo Pérez-Anzúrez, María Manuela Reyes-Estebanez, Pedro Mendoza de Gives.

**Resources:** Héctor Alejandro de la Crúz-Crúz, Rosa Isabel Higuera-Piedrahita.

**Supervision:** Yazmín Alcalá-Canto, Alejandro Zamilpa, Luis David Arango-de la Pava, Gerardo Ramírez-Rico.

**Validation:** Ana Yuridia Ocampo-Gutiérrez.

**Visualization:** Pedro Mendoza de Gives.

**Writing – original draft:** Héctor Alejandro de la Crúz-Crúz, Alejandro Zamilpa, Pedro Mendoza de Gives.

**Writing – review & editing:** Héctor Alejandro de la Crúz-Crúz, Rosa Isabel Higuera-Piedrahita, Yazmín Alcalá-Canto, Pedro Mendoza de Gives.

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
