## [Decision Letter · Decision Letter 0]

9 Nov 2025

PONE-D-25-51282Predatory activity and nematocidal compounds released into liquid culture filtrates as attack strategies of a Mexican strain of Arthrobotrys oligospora against Haemonchus contortus infective larvaePLOS ONE

Dear Dr. Higuera-Piedrahita,

Thank you for submitting your manuscript to PLOS ONE. After careful consideration, we feel that it has merit but does not fully meet PLOS ONE’s publication criteria as it currently stands. Therefore, we invite you to submit a revised version of the manuscript that addresses the points raised during the review process.

Overall, the reviewers considered the manuscript well-designed methodologically and with good prospects. However, changes and/or justifications are needed, particularly regarding the description of the statistical analysis of the results.==============================

We look forward to receiving your revised manuscript.

Kind regards,

Wesley Lyeverton Correia Ribeiro, Ph.D.

Academic Editor

PLOS ONE

Journal Requirements:

2. In your Methods section, please provide additional information regarding the permits you obtained for the work. Please ensure you have included the full name of the authority that approved the field site access and, if no permits were required, a brief statement explaining why

“To the Secretariat of Science, Humanities, Technology, and Innovation (SECIHTI) for the scholarship awarded to Professor Héctor Alejandro de la Cruz (registration number 713914). This research was also supported by SECIHTI (Frontier Sciences Project-2023, scholarship number CF-2023-I-2309). This work also received funding from UNAM DGAPA-PAPIIT 200324 "Molecular docking study of two lignans, 3-dimethoxy-isoguayacin and norisoguayacin, obtained from Artemisia cina against COX-2," and Cátedra FESC CI2428.”

6. We note that Figure 1 in your submission contain map images which may be copyrighted. All PLOS content is published under the Creative Commons Attribution License (CC BY 4.0), which means that the manuscript, images, and Supporting Information files will be freely available online, and any third party is permitted to access, download, copy, distribute, and use these materials in any way, even commercially, with proper attribution. For these reasons, we cannot publish previously copyrighted maps or satellite images created using proprietary data, such as Google software (Google Maps, Street View, and Earth). For more information, see our copyright guidelines: http://journals.plos.org/plosone/s/licenses-and-copyright.

Reviewers' comments:

Reviewer's Responses to Questions

**Comments to the Author**

1. Is the manuscript technically sound, and do the data support the conclusions?

Reviewer #1: Partly

Reviewer #2: Yes

2. Has the statistical analysis been performed appropriately and rigorously? 

Reviewer #1: I Don't Know

Reviewer #2: Yes

3. Have the authors made all data underlying the findings in their manuscript fully available?

Reviewer #1: Yes

Reviewer #2: Yes

4. Is the manuscript presented in an intelligible fashion and written in standard English?

Reviewer #1: Yes

Reviewer #2: Yes

5. Review Comments to the Author

Reviewer #1: General comment:

This manuscript presents solid, innovative, and highly interesting research. However, regarding the in vitro anthelmintic evaluation data of the fungus and its filtrate, it is necessary to clarify whether the assumptions underlying the statistical analysis were tested to ensure that the data met these requirements. Otherwise, the data should be transformed or analyzed using nonparametric methods to enhance the scientific rigor and validity of the conclusions.

Specific comments:

Key words: Please, instead the words “Haemonchus contortus” and “nematocidal activity” choose other two words that are not contained in your title.

Introduction:

In line 57, source 11 most not be included here because this article talks about nematophagous fungi extracts and not about plant extracts. Therefore, you can include this source in the part of the sentence where you talk about nematophagous fungi.

In line 66, I think it is important to mention the specific place in Mexico (city) where the fungi of the study were collected.

Figures:

Figure 1. Could you please substitute the figures A, B and C with other figures with better resolution?

Material and methods

All trade names must have the "®" mark, especially in the methodology section (lines 97, 110, etc.).

In line 132 please add any description of this H. contortus strain and a reference.

In line 139, it is important to express that the protocol was also “approved” and not only supervised by the Committee of Care and Use of Experimental Animals.

In line 144, please add the plate measurements.

In line 153, please add the reference Abbott formula.

Statistical design

Was it evaluated whether the data complied with the assumptions of the analysis of variance (homoscedasticity)? If yes, please add this information to this section, because this kind of data is frequently not normal, making it necessary to transform the data or use a non-parametric analysis.

In line 204, please put H. contortus in italics.

In the section Zymography assays, please add the ® mark to all the commercial names. Please check this in all the manuscript.

Results

Would it be possible to include images in Figures 2 and 4 with better resolution? Additionally, it would be interesting to include a scale showing the length in the images.

In table 1, This range corresponds to the minimum and maximum values observed or measured standard deviation.

In line 281, used italics for scientific names. Please ensure that all scientific names are consistently italicized throughout the manuscript.

In table 3, Since different letters were not observed in the means, there was no significant difference between the control and the treated group, right?

In table 4, I think it would be interesting to make statistical comparisons between the means of the different filtrate concentrations, and with the negative and positive controls used, could you add these results to this table?

Microscopic analysis:

These chances were also observed and lower concentration or not? were observed to a lesser degree? Please specify this in your results.

In line 132 the information cannot be read.

The information in the title of Figure 6 is very extensive. Is it possible to summarize it without omitting the essential information?

Discussion:

In line 364, add a reference to support your statement.

In line 387, please include against “infective larvae” of …the gastrointestinal parasitic nematode H. contortus, ….

Table 8 is not cited or mentioned in the discussion section and should be properly integrated into the text. In general, the number of tables in the discussion could be reduced, keeping only those essential to support or strengthen your arguments.

Do you think that the microscopic changes observed in the intestinal cells of the infective larvae could be due to the direct penetration of some of the metabolites through the intact cuticle and membranes, or rather to the prior enzymatic degradation of the cuticle by these proteolytic enzymes? Alternatively, do you think that the ingestion of these metabolites and fungal enzymes might also have contributed to the observed changes in the intestinal cells of the larvae? I believe it would be interesting to address this point in your discussion.

In line 510, it is important to indicate that the Internal Committee for the Care and Use of Experimental Animals (CICUAE-FESC) not only supervised but also approved the study.

Reviewer #2: 131 It does not specify whether the lamb was previously dewormed (anthelminthic treatment) or if it was evidenced to be free of gastrointestinal parasites.

143 What was the variability (standard deviation) of the mean values of recovered larvae with or without the action of the predatory fungus?

163 't'-test for small samples (discrimination by least significant difference?) please clarify, because in line 198, mean separation with Tukey's test was used for lethal activity. Was this design only for lethal activity or also for predatory activity?

254 What was the variability of the mean values (±s)?

General Comment: A very interesting study with several components, the results are promising regarding the biological control of gastrointestinal parasitism and contributing to the control of anthelmintic resistance. However, because this is an in vitro study with interesting results, the true in vivo activity of the fungus remains unknown.

As part of a doctoral thesis, it would be interesting to know if any phase or sequence for in vivo animal trials is being considered. It has been observed that many very promising in vitro results have not yielded the same outcome when applied in vivo, much less in field activity under real-world conditions. So far, the gap in the transition from in vitro to in vivo, with the same or better results has not been overcome. There are many more variables, but an approximation would be interesting to see.

6. PLOS authors have the option to publish the peer review history of their article (what does this mean? ). If published, this will include your full peer review and any attached files.

**Do you want your identity to be public for this peer review?** For information about this choice, including consent withdrawal, please see our Privacy Policy .

Reviewer #1: No

Reviewer #2: No

---

## [Author Response · Author response to Decision Letter 1]

1 Dec 2025

Cuautitlán Izcalli, November 29th, 2025

Dear

Emily J Chenette

Editor in Chief

Plos One

Dear Editor,

Here, we have prepared a list of the comments and suggestions made by the Reviewers and included the individual answers to every comment. Likewise, the corrections suggested by the Reviewer are being addressed in the new manuscript version. We hope this new version meets the standard quality required by your prestigious journal.

Journal Requirements:

2. In your Methods section, please provide additional information regarding the permits you obtained for the work. Please ensure you have included the full name of the authority that approved the field site access and, if no permits were required, a brief statement explaining why

Authors:

The permission for archeological zone was processed at National Service of Agrifood Health, Safety and Quality (SENASICA) for the for admission, sample collection, and authorization for publication of results. This is attached below. Line 80 -81.

Journal Requirements:

“To the Secretariat of Science, Humanities, Technology, and Innovation (SECIHTI) for the scholarship awarded to Professor Héctor Alejandro de la Cruz (registration number 713914). This research was also supported by SECIHTI (Frontier Sciences Project-2023, scholarship number CF-2023-I-2309). This work also received funding from UNAM DGAPA-PAPIIT 200324 "Molecular docking study of two lignans, 3-dimethoxy-isoguayacin and norisoguayacin, obtained from Artemisia cina against COX-2," and Cátedra FESC CI2428.”

Authors:

The sentence was added in financial disclosure, Line 504-505.

Journal Requirements:

Authors:

The data from the experiments are available

Journal Requirements:

Authors:

The original gel of the zymograms has been added in pdf format to the documents of this manuscript

Journal Requirements:

Authors:

The original zymogram gel were added in supplementary information

Journal Requirements:

6. We note that Figure 1 in your submission contain map images which may be copyrighted. All PLOS content is published under the Creative Commons Attribution License (CC BY 4.0), which means that the manuscript, images, and Supporting Information files will be freely available online, and any third party is permitted to access, download, copy, distribute, and use these materials in any way, even commercially, with proper attribution. For these reasons, we cannot publish previously copyrighted maps or satellite images created using proprietary data, such as Google software (Google Maps, Street View, and Earth). For more information, see our copyright guidelines: http://journals.plos.org/plosone/s/licenses-and-copyright.

Authors:

The figure 1 eas deleted from the manuscript

Journal Requirements:

Authors:

The references were added with Mendeley Program in Plos One style citation.

Reviewers' comments:

Key words: Please, instead the words “Haemonchus contortus” and “nematocidal activity” choose other two words that are not contained in your title.

Authors:

Two different key words were added

Reviewers' comments:

Introduction:

In line 57, source 11 most not be included here because this article talks about nematophagous fungi extracts and not about plant extracts. Therefore, you can include this source in the part of the sentence where you talk about nematophagous fungi.

Authors:

References were actualized. Source 11 were replaced by:

Ocampo-Gutiérrez AY, Hernández-Velázquez VM, Aguilar-Marcelino L, Cardoso-Taketa A, Zamilpa A, López-Arellano ME, et al. Morphological and molecular characterization, predatory behaviour and effect of organic extracts of four nematophagous fungi from Mexico. Fungal Ecol. 2021;49: 101004. doi:10.1016/j.funeco.2020.101004

Reviewers' comments:

In line 66, I think it is important to mention the specific place in Mexico (city) where the fungi of the study were collected.

Authors:

The specific soil sampling location from which the fungus was isolated is described in the methodology section under Sampling site. Line 76

Reviewers' comments:

Figures:

Figure 1. Could you please substitute the figures A, B and C with other figures with better resolution?

Authors:

Figure 1 was deleted

Reviewers' comments:

Material and methods

All trade names must have the "®" mark, especially in the methodology section (lines 97, 110, etc.).

Authors:

The mark was added in all methodology

Reviewers' comments:

In line 132 please add any description of this H. contortus strain and a reference.

Authors:

The information was added. Line 133- 135.

Reviewers' comments:

In line 139, it is important to express that the protocol was also “approved” and not only supervised by the Committee of Care and Use of Experimental Animals.

Authors:

The word was changed

Reviewers' comments:

In line 144, please add the plate measurements.

Authors:

The information was added in line 145

Reviewers' comments:

In line 153, please add the reference Abbott formula.

Authors:

The reference was added

Reviewers' comments:

Statistical design

Was it evaluated whether the data complied with the assumptions of the analysis of variance (homoscedasticity)? If yes, please add this information to this section, because this kind of data is frequently not normal, making it necessary to transform the data or use a non-parametric analysis.

Authors:

Due to the number of repetitions and the homogeneity of the data, it was decided to calculate only the standard deviation. This decision was made in consultation with Dr. Agustín Olmedo Juárez, the statistician on the research team.

Reviewers' comments:

In line 204, please put H. contortus in italics.

Authors:

The word was edited

Reviewers' comments:

In the section Zymography assays, please add the ® mark to all the commercial names. Please check this in all the manuscript.

Authors:

The suggestion was added

Reviewers' comments:

Results

Would it be possible to include images in Figures 2 and 4 with better resolution? Additionally, it would be interesting to include a scale showing the length in the images.

Authors:

The scale was added

Reviewers' comments:

In table 1, This range corresponds to the minimum and maximum values observed or measured standard deviation.

Authors:

The range corresponds to the standard deviation

Reviewers' comments:

In line 281, used italics for scientific names. Please ensure that all scientific names are consistently italicized throughout the manuscript.

Authors:

The word is in italics now

Reviewers' comments:

In table 3, Since different letters were not observed in the means, there was no significant difference between the control and the treated group, right?

Authors:

There was statistical differences, the letters were added

Reviewers' comments:

In table 4, I think it would be interesting to make statistical comparisons between the means of the different filtrate concentrations, and with the negative and positive controls used, could you add these results to this table?

Authors:

The table 4 were re-structurated with controls and statistical comparison

Reviewers' comments:

Microscopic analysis:

These chances were also observed and lower concentration or not? were observed to a lesser degree? Please specify this in your results.

Authors:

The specifications were added. Line 291.

Reviewers' comments:

In line 132 the information cannot be read.

Authors:

The paragraph was rewritten

Reviewers' comments:

The information in the title of Figure 6 is very extensive. Is it possible to summarize it without omitting the essential information?

Authors:

The information readjusted for minus words.

Reviewers' comments:

Discussion:

In line 364, add a reference to support your statement.

Authors:

The reference was added

Reviewers' comments:

In line 387, please include against “infective larvae” of …the gastrointestinal parasitic nematode H. contortus, ….

Authors:

The phase was rewritten. Line 372

Reviewers' comments:

Table 8 is not cited or mentioned in the discussion section and should be properly integrated into the text.

Authors:

The information was integrated. Line 404 -405.

Reviewers' comments:

In general, the number of tables in the discussion could be reduced, keeping only those essential to support or strengthen your arguments.

Authors:

The authors believe that although the information in tables is extensive, it is necessary to fully understand the experiment and for its future replication by other academic teams.

Reviewers' comments:

Do you think that the microscopic changes observed in the intestinal cells of the infective larvae could be due to the direct penetration of some of the metabolites through the intact cuticle and membranes, or rather to the prior enzymatic degradation of the cuticle by these proteolytic enzymes? Alternatively, do you think that the ingestion of these metabolites and fungal enzymes might also have contributed to the observed changes in the intestinal cells of the larvae? I believe it would be interesting to address this point in your discussion.

Authors:

The paragraph with explanation was added in discussion section. Line 420 - 434

Reviewers' comments:

In line 510, it is important to indicate that the Internal Committee for the Care and Use of Experimental Animals (CICUAE-FESC) not only supervised but also approved the study.

Authors:

The word was changed

Reviewers' comments:

131 It does not specify whether the lamb was previously dewormed (anthelminthic treatment) or if it was evidenced to be free of gastrointestinal parasites.

Authors:

The specification was added. Line 130

Reviewers' comments:

143 What was the variability (standard deviation) of the mean values of recovered larvae with or without the action of the predatory fungus?

Authors:

The declaration was included in line 152

Reviewers' comments:

163 't'-test for small samples (discrimination by least significant difference?) please clarify, because in line 198, mean separation with Tukey's test was used for lethal activity. Was this design only for lethal activity or also for predatory activity?

Authors:

The declaration was added in line 165

Reviewers' comments:

254 What was the variability of the mean values (±s)?

Authors:

Standard deviation was added. Line 249

Reviewers' comments:

General Comment: A very interesting study with several components, the results are promising regarding the biological control of gastrointestinal parasitism and contributing to the control of anthelmintic resistance. However, because this is an in vitro study with interesting results, the true in vivo activity of the fungus remains unknown.

Authors:

There is currently a manuscript in the process of being published that demonstrates the activity of the fungus in rearing lambs infected with Haemonchus contortus.

Similarly, there is a 2025 publication on the activity of the nematophagous fungus on first-calf ewes infected with the same parasite.

de la Crúz-Crúz, H. A., Higuera-Piedrahita, R. I., Zamilpa, A., Alcalá-Canto, Y., Ocampo-Gutiérrez, A. Y., Arango-de la Pava, L. D., López-Arellano, M. E., Hernandez-Patlan, D., Cuéllar-Ordaz, J. A., & Mendoza-de Gives, P. (2025). Using an Aqueous

---

## [Editor Report · Decision Letter 1]

7 Jan 2026

Predatory activity and nematocidal compounds released into liquid culture filtrates as attack strategies of a Mexican strain of Arthrobotrys oligospora against Haemonchus contortus infective larvae

PONE-D-25-51282R1

Dear Dr. Higuera-Piedrahita,

We’re pleased to inform you that your manuscript has been judged scientifically suitable for publication and will be formally accepted for publication once it meets all outstanding technical requirements.

Kind regards,

Wesley Lyeverton Correia Ribeiro, Ph.D.

Academic Editor

PLOS One
---

## [Editor Report · Acceptance letter]

PONE-D-25-51282R1

PLOS One

Dear Dr. Higuera-Piedrahita,

I'm pleased to inform you that your manuscript has been deemed suitable for publication in PLOS One. Congratulations! Your manuscript is now being handed over to our production team.

Kind regards,

on behalf of

Dr. Wesley Lyeverton Correia Ribeiro

Academic Editor

PLOS One